# Dissipation Theory-Based Ecological Protection and Restoration Scheme Construction for Reclamation Projects and Adjacent Marine Ecosystems

**DOI:** 10.3390/ijerph16214303

**Published:** 2019-11-05

**Authors:** Faming Huang, Yanhong Lin, Rongrong Zhao, Xuan Qin, Qiuming Chen, Jie Lin

**Affiliations:** 1Third Institute of Oceanography, Ministry of Natural Resources, Xiamen 361005, China; linyanhong@tio.org.cn (Y.L.); zhaorongrong@tio.org.cn(R.Z.); chenqiuming@tio.org.cn(Q.C.); linjie@tio.org.cn(J.L.); 2College of Environment and Ecology, Xiamen University, Xiamen 361005, China; linyanhong@tio.org.cn; 3Xiamen Branch of Tianjin Urban Planning and Design Institute, Xiamen 3611005, China; lydia710_qin@hotmail.com

**Keywords:** ecological restoration, dissipative structure, entropy change, analytic hierarchy process

## Abstract

According to the 2017 results of the Special Inspector of Sea Reclamation, a substantial number of idle reclamation zones existed in 11 provinces (cities) along the coast of China. To improve the protection level of coastal wetlands and strictly control reclamation activities, it is necessary to carry out ecological restoration of reclamation projects and adjacent marine ecosystems. The characteristics of Guanghai Bay and its reclamation project are typical in China’s coastal areas, making it an optimal representative site for this study. The dissipative structure and entropy theory was used to analyze ecological problems and environmental threats. The analytic hierarchy process was applied to determine the order of the negative entropy flow importance. The entropy increase and decrease mechanism was used to determine an ecological protection and restoration scheme for the reclamation, including the reclamation of wetland resource restoration, shoreline landscape restoration, environmental pollution control, and marine biological resource restoration. Finally, based on system logic, a typical ecological restoration system was constructed east of Guanghai Bay, with the mangrove wetland area as the model in the north and the artificial sandbeach recreation area as the focus in the south.

## 1. Introduction

Coastal wetlands (including beaches, estuaries, shallow seas, mangroves, and coral reefs) are precious wetland resources with significant ecological functions, which provide important habitats, breeding sites, and migration stations for birds [1]. To provide sufficient land space for the socio-economic development, China’s coastal cities have carried out long-term and large-scale reclamation activities from 2002 to 2018. Due to this, its coastal wetlands have substantially decreased, and the natural coastline has sharply reduced, damaging marine and terrestrial ecosystems [2]. However, according to the survey results of the Special Inspector of Sea Reclamation in 2017 [3], all provinces (cities) in China’s coastal areas face the problem of sea-reclamation projects remaining largely idle [4,5,6,7,8,9,10,11]. To improve the protection level of coastal wetlands and control reclamation activities, it is necessary to ecologically assess the reclamation projects as soon as possible and propose reasonable and feasible ecological restoration measures for reclamation projects and their adjacent marine ecosystems, thereby minimizing the impacts on ocean hydrodynamics and biodiversity [12,13,14].

Reclamation projects and their adjacent marine ecosystems possess typical features of a dissipative structure, which include (1) an open system, (2) nonequilibrium, (3) nonlinear interaction, and (4) fluctuation. A dissipative structure is a kind of dynamic ordered structure state, and change processes can be demonstrated based on the entropy change of the dissipative structural system [15]. Entropy is a state parameter that measures the degree of order of the system. A higher entropy value indicates a more disordered system and a lower value indicates a more ordered system [16,17]. In recent years, a combination of the dissipative structure theory and entropy change method has been developed to evaluate and improve various ecosystems. Zhang et al. (2006) developed an indicator system based on the dissipative structure theory and a model based on information entropy to estimate various flows in an urban ecosystem [18]. Wu et al. (2013) proposed the maintenance and orderly development of island ecological–environmental systems by increasing negative entropy flow [15]. Ludovisi et al. (2014) studied the directional role of several entropy-based indicators (e.g., structural information, specific entropy production, and the eco-exergy index) in the ecological succession of eutrophication [17]. Di and Han (2014) used the information entropy method to judge the sustainable development capability of marine ecosystems at the national level [19]. Wang et al. (2018) constructed the “Marine Ecosystem Evolution Index System” based on the information entropy theory [20]. Xu et al. (2019) proposed the modified entropy weight of the AHP model, which can be used in the construction of regional information of ecological environments [21]. Zhou (2019) analyzed the relationship of ecological compensation based on the principle of maximum entropy and designed the calculation method of ecological compensation standard for adjacent administrative districts [22]. These studies showed that the combination of the dissipative structure theory and entropy change method for studying the evolution and sustainable development of urban ecosystems has been extremely common. However, this combination has rarely been applied for the construction of ecological restoration schemes for reclamation zones and adjacent marine ecosystems.

Therefore, the objectives of this study are to analyze ecological problems and environmental threats faced by typical reclamation projects and their adjacent marine ecosystems, based on the theory of dissipative structure and the entropy theory, and then to utilize the mechanism of entropy increase and decrease to determine ecological protection and restoration plans. In addition, the characteristics of Guanghai Bay and the reclamation project in it are typical at China’s coastal areas, making it a representative site for analyzing the ecological problems and environmental threats arising from reclamation based on the dissipative structure theory and entropy change. Finally, a restoration scheme for wetland resource restoration, shoreline landscape restoration, environmental pollution control, and marine biological resource restoration was proposed, and a typical ecological restoration system considering the mangrove wetland area in the north and the artificial sandbeach recreation area in the south was constructed. The results of this paper can be used as a reference for formulating ecological protection and restoration schemes of other reclamation projects and their adjacent marine ecosystems.

## 2. Materials and Methods

### 2.1. Study Area

Guanghai Bay is located at the south sea area of Jiangmen City, Guangdong Province (Figure 1). The east, north, and west sides are connected to the open inland hinterland, while facing the sea in the south. Guanghai Bay is a typical semi-enclosed bay. Currently, there is only one reclamation project being conducted in the Bay, which has formed a land area.

The east side of Guanghai Bay and the land behind it are currently planned to be reformed as Guanghai harbor industrial park, whose harbor industries include marine engineering equipment manufacturing and science and education research and development. The planned land area is approximately 21.2 km^2^, of which approximately 10.4 km^2^ needs to be obtained via reclamation activities. Therefore, the Guanghaiwan reclamation project was launched in Guanghai Bay to solve the problem of land shortage in Guanghaiwan Industrial Park. Currently, only approximately 6.6 km^2^ of these reclamation projects has been completed (divided into areas C, D, and E from the north to the south). As the Chinese government has formulated the strictest reclamation policy from 2018, subsequent reclamation plans have been shelved. The scale of the 6.6 km^2^ of reclamation and adjacent marine ecosystem were consider as the study area, and case studies were conducted based on the assumption that subsequent reclamation projects would not be implemented.

### 2.2. Research Methods

#### 2.2.1. Dissipative Structure and Establishment of Entropy Model

Considering the four characteristics of a dissipative structure, reclamation projects and their adjacent marine ecosystems are a typical dissipative structure. First, they are open systems [23], where animals, plants, and microorganisms are constantly exchanged with the surrounding environment for material and energy, and they are significantly affected by environmental factors. Second, there are spatial differences, functional differences, and four seasons of transformation in the system, which results in the system being far from the equilibrium state [24]. Third, the dissipative system is a nonlinear dynamic process. The reclamation project and its adjacent ecosystem display a positive and negative feedback mechanism, and the system is evolved and updated according to nonlinear laws. Fourth, as a typical complex ecosystem, the reclamation project and its adjacent ecosystem are likely to suffer from random and uncertain fluctuations, such as human interference and natural evolution. This causes the system to deviate from its normal state, thereby forming fluctuations, and to rely on its self-organizing abilities to adjust the structure and functions, finally forming a new dissipative structure.

In 1948, Shannon proposed the concept of “information entropy”, which is used to represent the degree of disorder of the system and interpret the evolution direction of the system [15,25,26]. A higher entropy value (*S*) indicates a more disordered system and a lower value indicates a more ordered system [16,17,27]. The entropy change (Δ*S*) of an open system consists of two parts: the entropy increase (Δ*S*i) caused by the internal irreversible process of the system (its value is always greater than zero) and the entropy reduction (Δ*S*e) caused by the exchange of matter or energy with the outside world (its value may be positive, negative, or zero). That is, the total entropy of the open system is the sum of the entropy increase and decrease, and the total entropy (Δ*S*) of the open system is determined as follows [28]:(1)ΔS=ΔSi+ΔSe

According to Equation (1), the positive and negative values of the total entropy change (Δ*S*) of the system are dependent on the scale and magnitude of the increase and decrease in the entropy. That is, the system can only progress toward ordering when Δ*S*e < 0 and |Δ*S*e| > Δ*S*i. At this time, Δ*S* = Δ*S*i + Δ*S*e < 0.

A nonlinear open system that is far from the equilibrium state can only self-organize if the system draws a sufficient negative entropy from the outside to reduce the total entropy. Therefore, to develop in a dynamic and orderly manner, the system must acquire a large amount of negative entropy by exchanging matter and energy with the outside world.

As it is a dissipative-structure system, the entropy values inside reclamation projects and their adjacent marine ecosystems change constantly during the development process. As illustrated in Figure 2, when entropy reduction dominates, the system efficiency will gradually increase, and the system will change from disordered to ordered (segment AB). As time increases, the system entropy increases continuously. When the entropy increases and plays the dominant role, the system efficiency decreases and the system will change from ordered to disordered (segment BC). Here, if the system does not introduce high-quality and effective negative entropy reduction in time, the system will eventually be eliminated (segment CD). If appropriate amounts of high-quality negative entropy reduction of matter, information, and energy, among others, are introduced in time to break the balance, the system development will regain its vitality (segment CE). The increase and decrease in the entropy of the system exists in a state of confrontation and growth [29]. Therefore, the entropy increase and decrease factors of the reclamation projects and their adjacent marine ecosystems must be analyzed under different stages and conditions.

#### 2.2.2. Analytic Hierarchy Process

Analytic hierarchy process (AHP) is a hierarchical weighted decision analysis method proposed by Professor T.L. Satty of the University of Pittsburgh [31,32,33]. It decomposes the relevant elements of a decision-making problem into goals, criteria, and programs, and accordingly conducts qualitative and quantitative analyses. The application of AHP involves four main steps:(1)Establishment of a hierarchical structure model:

First, the objectives of the decision, factors considered (evaluation criteria) and decision objects (action plans), are divided into the top layer (target layer-AW), middle layer (guidelines layer-BW), and bottom layer (measures layer-CW), according to their mutual relationship.
(2)Construction of a comparison discriminant matrix:

Once the hierarchy is established, the evaluator or experts score the factors according to their experience, starting from the first criterion level. The importance of the different factors of each layer relative to other factors is gradually weighted, generally through a pair-wise comparison method. The comparative relationship of each element is summarized in a comparison judgment matrix A, as follows:
(2)$$A=\left[\begin{array}{cccc} a_{11} & a_{12} & \cdots & a_{1n}\\ a_{21} & a_{22} & \cdots & a_{2n}\\ \vdots & \vdots & \vdots & \vdots \\ a_{n1} & a_{n2} & \cdots & a_{nn} \end{array}\right]$$
where *a_ij_* generally takes a positive integer from 1 to 9 (known as the scale) and its reciprocal; that is, if factor *i* is compared with factor *j* to obtain *a_ij_*, factor *j* is compared with factor *i* by using 1/*a_ij_* (where *i* and *j* = 1, 2, ···, *n*). The rule for establishing the value of *a_ij_* is indicated in Table 1 [34]. Then, the hierarchical single-sorted weight vector (*W*) and maximum eigenvalue (λ_max_) can be calculated using matrix A.
(3)Consistency test:

When constructing a comparison judgment matrix, the judgment does not require complete consistency; however, the direction should be substantially uniform. Therefore, a consistency test is required for each level of single-criterion ordering.

Let A be an *n*-order positive cross-reverse matrix, where the consistency standard (*CI*) is defined as
(3)$$CI=\frac{\lambda_{\max}-n}{n-1},$$
where λ_max_ is the maximum eigenvalue of matrix A and *CI* is the quantitative standard for measuring the inconsistency degree.

When the maximum eigenvalue of matrix A is slightly larger than *n*, A is said to exhibit satisfactory consistency. The following evaluation method was adopted in this study: a fixed *n* is used to construct a positive reciprocal matrix A = (*a_ij_*)*_n_* randomly, where *a_ij_* ranges from 1, 2, 3, …, 9, 1/2, 1/3, …, 1/9 out of the total of 17 numbers. Such a positive cross-reverse matrix A is the most inconsistent. The maximum characteristic λ_max_ of the above-mentioned random judgment matrix is calculated 1000 times, and the given random consistency index (*RI*) value is presented in Table 2.

In Table 2, for *n* = 1 and 2, *RI* = 0 because the first- and second-order judgment matrices are always consistent. When *n* ≥ 3, the consistency ratio (*CR*) is determined as follows:(4)CR=CI/RI.

When *CR* < 0.1, the consistency of the comparison judgment matrix is acceptable; otherwise, the judgment matrix should be appropriately corrected.

## 3. Results

### 3.1. Main Form of Entropy Increase of Reclamation Project and Adjacent Marine Ecological Environment

#### 3.1.1. Entropy Increases in Resources: Reduced Wetland Area and Loss of Biological Resources

The reclamation project is approximately 6.6 km^2^, and it is located within the 0 m of water depth (Figure 3), which belongs to the coastal wetland (within the 6-m of water depth) in a broad sense. Development and utilization methods, including the extension toward the sea level, cutting off the bend to make it straight, and other simple types, which are adopted in this reclamation project, resulted in a series of problems such as shortening of the natural shoreline, a loss of wetland resources, and a reduction in marine biodiversity. The occupation of wetlands transforms the original sea area into land and changes the natural attributes and ecological environment of the original coast, intertidal zone, and sea area. This destroys the ecological service functions of wetlands, with wetland organisms losing their living and reproductive space. Moreover, with the destruction of marine biological resources, their output is reduced, leading to the loss of wetland habitats; this has a negative impact on the population composition and the spatio-temporal distribution of benthic animals and birds. During the process of construction behaviors, such as riprap, blasting to squeeze out silt, push-fill overflow in the land area, and permanent occupation of the seabed subsoil through sea reclamation, has resulted in the loss of marine resources. According to a July 2017 survey of bio-ecological and fishery resources in the Guanghai Bay conducted by the Ocean Monitoring and Testing Center of the Ocean University of China, this reclamation project resulted in a loss of approximately 10 t of plankton, approximately 17 t of swimming creatures, 1.2 × 10^8^ grains of fish eggs, 9.0 × 10^7^ tails of larvae, approximately 320 t of benthic organisms, approximately 800 t of intertidal organisms, 5.4 t of fish, and 1.8 t of crustaceans.

#### 3.1.2. Entropy Increases in Environment: Soil Erosion, Near-Shore Pollution, and Reduced Environmental Capacity

The reclamation materials in this reclamation project contained high soil content and formed a land area over more than 10 years (Figure 4a,b). At present, the soil is still directly exposed or only grows sporadic pioneer plants, mostly herbaceous and vine plants (Figure 4c). The soil in the reclamation area is soft and does not comprise a large-scale vegetal cover. Under the conditions of hydraulics, gravity, and wind erosion, and particularly high-intensity rainstorm conditions, soil erosion occurs easily. Soil erosion destroys land resources and causes a large amount of sediment to flow into rivers. Increased sediment inflow into rivers can easily lead to siltation of the surrounding rivers, such as the Dama and Xiaoma Rivers, and could easily cause flooding disasters, resulting in a decrease in the quality of the ecological environment and pollution of the surrounding water bodies.

Simultaneously, the sea reclamation has intensified the near-shore environmental pollution of Guanghai Bay, and the permanent change in the natural attributes of the marine ecosystem due to sea reclamation has led to the disappearance of the environmental capacity of this part of the ocean. According to the results of a marine environmental quality survey conducted in July 2017, the loss in the environmental capacity of chemical oxygen demand caused by sea reclamation in the reclamation area was 2.77 t/a, while the capacity loss of total nitrogen (TN) and total phosphorus (TP) caused by the occupied sea area was 7.99 t/a.

#### 3.1.3. Entropy Increases in Landscape: Poor Public-Service Function and Landscape Effect

The rigid flat design and single hard revetment form had led to poor public-service function, and landscape effect. The layout of the reclamation, particularly in area D in the reclamation area, is stiff, forming a plurality of near 90° revetment corners (Figure 4d). The rivers, including the Dama and Xiaoma rivers between the reclamation areas, are straight, which reduces the diversity of the ecological environment of the river water and leads to the degradation of the river ecosystem [35]. This, in turn, would lead to the entrainment of pollutants in upstream waters that directly enter the reclamation areas, and the organisms that could take refuge in the slow-flow area of the bay during floods would not survive. The erosion of the tidal current on such a revetment is more severe than that of a meandering revetment, thus affecting the revetment stability. The revetments of the reclamation area are illustrated in Figure 4e–g and are made up of hard stone blocks; the hard sea reclamation revetment form cuts off biological exchange between the water body and land. Apart from a small area of mangroves that has been planted in front of the revetment, coastal plants are scarce. The revetment has not been implemented with a reasonable and attractive hydrophilic design. The inert materials were simply stacked and treated to form a large piece of bare soil, and the landscape effect on the revetment is poor. The resource advantages of the coastal wetlands are not effectively reflected, with lack of space sharing, public hydrophilic zones, and public service functions [36,37].

### 3.2. Relative Importance of Calculating Negative Entropy Flow by Analytic Hierarchy Process (AHP)

The maintaining of the dynamic and orderly development of the system does not negatively affect the status quo but constantly breaks the balance, with external matter and energy continually being absorbed to dissipate the internal low quality and high entropy to maintain an orderly structure. The analysis of the reclamation project and the entropy increase of its adjacent marine ecological environment showed that the reclamation projects and their adjacent marine ecosystems rely solely on their self-organization capabilities for structure and function owing to problems with resources, environment, and landscape ecology. It is difficult for the adjustment to form a new dissipative structure, and a negative entropy input outside the system is required. Combining the requirements for ecological protection and restoration of reclamation projects in the “Technical Guidelines for the Preparation of Ecological Protection and Restoration Schemes for Reclamation Projects (Trial),” the restoration of marine biological resources, coastal wetland restoration, shoreline restoration, and pollution prevention measures may be adopted to introduce negative entropy flow.

To evaluate the importance of the proposed restoration measures, a hierarchical structure model was constructed using the AHP (Figure 5), and its weight was calculated.

Thereafter, a comparative discriminant matrix was constructed, and after consultation, the importance of the different factors of each layer relative to the upper layer factor was gradually weighted. The comparison discriminant matrix A is formulated as follows:(5)$$\mathrm{AW}=\left[\begin{array}{cccc} {\mathrm{BW}}_{11} & {\mathrm{BW}}_{12} & {\mathrm{BW}}_{13} & {\mathrm{BW}}_{14}\\ {\mathrm{BW}}_{21} & {\mathrm{BW}}_{22} & {\mathrm{BW}}_{23} & {\mathrm{BW}}_{24}\\ {\mathrm{BW}}_{31} & {\mathrm{BW}}_{32} & {\mathrm{BW}}_{33} & {\mathrm{BW}}_{34}\\ {\mathrm{BW}}_{41} & {\mathrm{BW}}_{42} & {\mathrm{BW}}_{43} & {\mathrm{BW}}_{44} \end{array}\right]=\left[\begin{array}{cccc} 1 & 2 & 5 & 3\\ 1/2 & 1 & 3 & 2\\ 1/5 & 1/3 & 1 & 1/2\\ 1/3 & 1/2 & 2 & 1 \end{array}\right]$$
(6)BW1=CWa11CWa12CWa13CWa14CWa15CWa21CWa22CWa23CWa24CWa25CWa31CWa32CWa33CWa34CWa35CWa41CWa42CWa43CWa44CWa45CWa51CWa52CWa53CWa54CWa55=123691/212571/31/21251/61/51/2121/91/71/51/21
(7)BW2=CWb11CWb11CWb11CWb11CWb21CWb22CWb23CWb24CWb31CWb32CWb33CWb34CWb41CWb42CWb43CWb44=12351/21241/31/2121/51/41/21
(8)$${\mathrm{BW}}_{3}=\left[\begin{array}{ccc} C_{c}W_{11} & C_{c}W_{12} & C_{c}W_{13}\\ C_{c}W_{21} & C_{c}W_{22} & C_{c}W_{23}\\ C_{c}W_{31} & C_{c}W_{32} & C_{c}W_{33} \end{array}\right]=\left[\begin{array}{ccc} 1 & 2 & 3\\ 1/2 & 1 & 2\\ 1/3 & 1/2 & 1 \end{array}\right]$$
(9)$${\mathrm{BW}}_{4}=\left[\begin{array}{ccc} C_{d}W_{11} & C_{d}W_{12} & C_{d}W_{13}\\ C_{d}W_{21} & C_{d}W_{22} & C_{d}W_{23}\\ C_{d}W_{31} & C_{d}W_{32} & C_{d}W_{33} \end{array}\right]=\left[\begin{array}{ccc} 1 & 2 & 3\\ 1/2 & 1 & 2\\ 1/3 & 1/2 & 1 \end{array}\right]$$
where BW_1_ is coastal wetland restoration, BW_2_ is shoreline restoration, BW_3_ is marine biological resource restoration and BW_4_ is pollution prevention.

In the third step, the weight vector was calculated and tested once (Table 3).

Table 3 indicates that the consistency of the comparison judgment matrix is acceptable. The ecological restoration measures to be taken are sorted according to importance. From the perspective of the guidelines, the following order is considered: coastal wetland restoration (BW_1_) > shoreline restoration (BW_2_) > pollution prevention (BW_4_) > marine biological resource restoration (BW_3_).

From the perspective of measures, the importance is as follows:(1)BW_1_: water system recovery > mangrove planting > returning beach from fish farming > returning wetland from farmland > control to alien species;(2)BW_2_: sandbeach conservation > artificial ecological revetment > vegetation planting > promoting siltation and maintaining siltbeach;(3)BW_3_: proliferation and release marine life > artificial fish reef > large algae cultivation;(4)BW_4_: reclaimed water reuse > sewage centralized treatment > sea-drifting garbage collection.

### 3.3. Main Ecological Restoration Scheme for Reclamation Projects and Adjacent Marine Ecosystems

According to the entropy change model and the degree of importance calculated via the analytic hierarchy process, ecological restoration schemes should be proposed in the following order: ecological restoration of the wetland system, ecological seawall construction, pollution prevention, and restoration of marine biological resources.

#### 3.3.1. Negative Entropy Flow of Resources: Ecological Restoration of Wetland Systems

The restoration of water system and mangrove plantation with the highest importance value were selected as restoration schemes of the wetland space (Figure 6a).
(1)Restoration of the water system:

Widen the width of the Dama and Xiaoma Rivers to ensure flood discharge and drainage. Green spaces, including a green gallery, green ring, and green heart, should be constructed at various areas, including sheltering belts on both sides of the coastal river channel (Figure 6b), to build a complete and coherent green space system and increase landscape, leisure, entertainment, and ecological functions.
(2)Mangrove ecological wetland area:

The entrance to the sea outside the Dama River and the north side of area C were selected as areas for mangrove planting (Figure 6c–e) because the water at these locations is relatively open, which facilitates the silt deposition. Furthermore, a small number of mangroves with eggplant and white bone soil were successfully planted at the north side of area C. The total area of the mangrove ecological restoration is approximately 270,000 m^2^.

In addition, there is still silt formed by riprap during revetment construction at the west side of area C. Therefore, mangroves can be planted on the outside, semi mangroves and herbaceous salt tolerant plants can be planted on the inside, and vegetation can be used to slow down wave erosion. This would improve the biodiversity in the coastal wetland and restore the habitats of foraging seabirds and beach organisms at these areas.

#### 3.3.2. Negative Entropy Flow of Landscape: Ecological Seawall Construction to Improve Landscapes

The sandbeach conservation and artificial ecological revetment (including vegetation planting) with the highest importance values were selected as measures for ecological shoreline restoration (Figure 7).
(1)Sandbeach shoreline:

Artificial sandbeach restoration on the south side of area D would protect and enhance the connection between the coastal city and the coastal space, providing space for public activities. The shoreline length of the restored sandbeach is approximately 1260 m, forming a sandbeach shoulder width of about 40–60 m and a sandbeach area of approximately 120,000 m^2^. It is estimated that the amount of sand to be backfilled is approximately 250,000 m^3^. The south end of the sandbeach is provided with a sand retaining (diversion) dike with a length of approximately 150 m.
(2)Artificial ecological revetment and vegetation planting

The shoreline have the form of slope revetment and vertical revetment. Based on the existing embankment structure in the reclamation area, the former is arranged in the northwest, west, and south of area C, as well as the northwest, west, and south of area D; the latter is mainly arranged in the west of area E (Figure 7).

Sloping shoreline: Vines are proposed to be planted above the masonry revetment to allow them to cover the revetment. In the gap of the masonry face, the revetment surface would be filled with fillers with a high efficiency of water absorption and nutrition, in which herbs or vines could be planted to form a plant surface covering the masonry face.

Upright shoreline: The greening center will mainly be the space enclosed by the inner side of the embankment and road. This offers anti-wave, windproof, and landscape effects, and will exhibit sound ecological stability. This is a complex coastal ecological protection system with a high ecological service function.

#### 3.3.3. Environmental Negative Entropy Flow: Pollution Prevention

The reclaimed water reuse and sewage centralized treatment with the highest importance value were selected as measures of sewage discharge and control.

Industrial wastewater and domestic sewage recycling should be promoted in the reclamation region, which should be combined with the construction of artificial ecological wetland and water-system. If sewage and wastewater must be discharged to the sea, they should be treated using the highest standard treatment method according to the water quality requirements of the marine functional area and total pollutant control requirements. Moreover, centralized drainage, offshore drainage, and ecological discharge should be adopted as much as possible. Ecological discharge encourages the discharge of sewage after its ecological treatment and fully exerts the repurification effect of ecological projects such as constructed wetlands.

#### 3.3.4. Bio-Ecological Negative Entropy Flow: Restoration of Marine Living Resources

The proliferation and release of marine life with the highest importance value were selected as measures of the restoration of marine living resources.

According to the local main biological species, the most damaged species and the species with the most obvious effects of proliferation and release, the main proliferation and release species are Black snappers, Yellow fin snappers and Penaeus monodon. Most importantly, the amount of proliferation and release of marine life must not be less than the loss amount.

## 4. Discussion

Due to the increasingly intensive sea use activities in the coastal zone, ecological protection and restoration work has been gradually taken seriously by regions and governments around the world to address resource loss and functional decline in many coastal ecosystems [38]. However, the current ecological protection and restoration works are mostly limited to a single level of technical restoration [39], such as only mangrove planting [40] or only coral reef restoration [41], but the ecosystem needs to be considered as a whole [42]. The single level of restoration hinders people from using the system concept to realize overall protection and restoration. However, it’s necessary that all factors of the restoration project should be connected in series to form an independent, interconnected, and mutually dependent whole, according to the different ecological restoration objects, varying damage degrees, and different stages. In addition, the ecosystem structure must be reconstructed or repaired. Social, economic, environmental, and other factors are combined with the superposition effect of point, line, and surface repairs [41]. Therefore, in this study, based on the current water system and natural environment resource background conditions, we considered the ranking of the importance of the negative entropy reduction calculated through AHP, and reorganized the ecological pattern of the Guanghai Bay reclamation project and its adjacent marine ecosystem. Through the integration management concepts of the mountain, water, forest, field, lake and grassland systems, a typical ecological restoration system was constructed at the east of Guanghai Bay, with the mangrove wetland area as the model in the north and the artificial sandbeach recreation area as the focus in the south (Figure 8).

Entropy increase caused by problems in resources, ecology, environment, and landscape results in the disordered development of reclamation projects and their adjacent marine ecosystems. By introducing external negative entropy flow, the overall entropy reduction of the system can be realized. The reclamation projects and their adjacent marine ecosystems are significantly different from the average value under the action of entropy increase and decrease, thereby promoting their orderly and dynamic development (Figure 9) [16]. However, this study only qualitatively analyzed the entropy model of the reclamation projects and their adjacent marine ecosystems and supplemented the relative importance ranking of the negative entropy flow using AHP. The construction of a quantitative analysis model for the entropy would be considered in the future research. The change in the value of entropy can further clarify the evolution mechanism of the reclamation projects and their adjacent marine ecosystems to develop a more targeted ecological protection and restoration program.

## 5. Conclusions

In this study, the dissipative structure and entropy theory was used to analyze ecological problems and environmental threats faced in the Guanghai Bay and the restoration. The problems associated with resources, ecology, environment, and landscape are addressed. The AHP was applied to determine the order of negative entropy flow importance. Finally, the entropy increase and decrease mechanism was used to determine the ecological protection and restoration scheme for reclamation, including the reclamation of wetland resource restoration, shoreline landscape restoration, environmental pollution control, and marine biological resource restoration. The conclusions provide a reference for formulating ecological protection and restoration schemes for reclamation projects and their adjacent marine ecosystems in other regions, as well as for promoting the construction of a marine ecological civilization in coastal areas.

## Figures and Tables

**Figure 1 ijerph-16-04303-f001:**
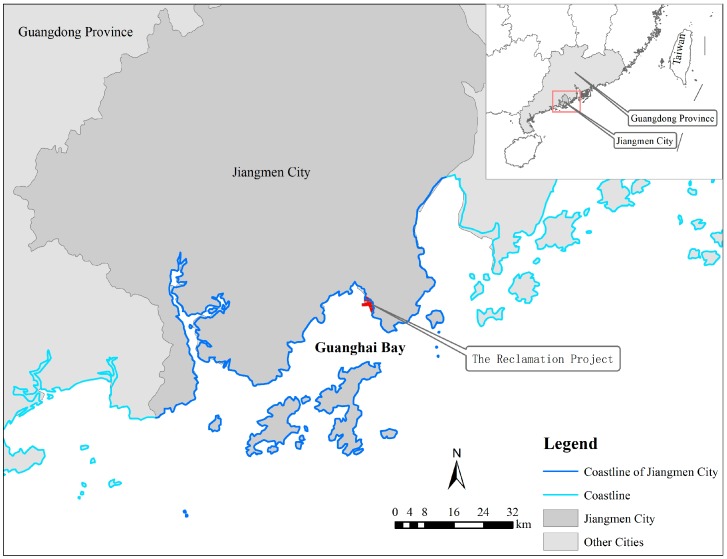
Location of Guanghai Bay in Guangdong Province, China.

**Figure 2 ijerph-16-04303-f002:**
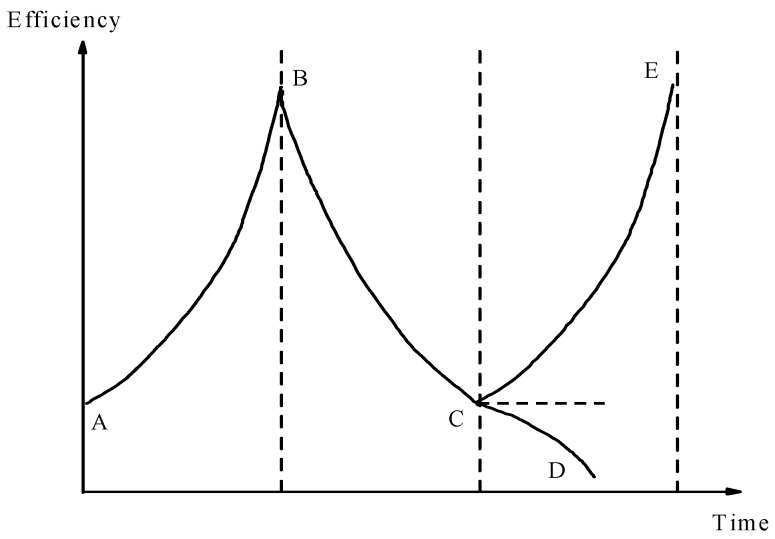
The entropy change of reclamation projects and their adjacent marine ecosystems with time changes [30].

**Figure 3 ijerph-16-04303-f003:**
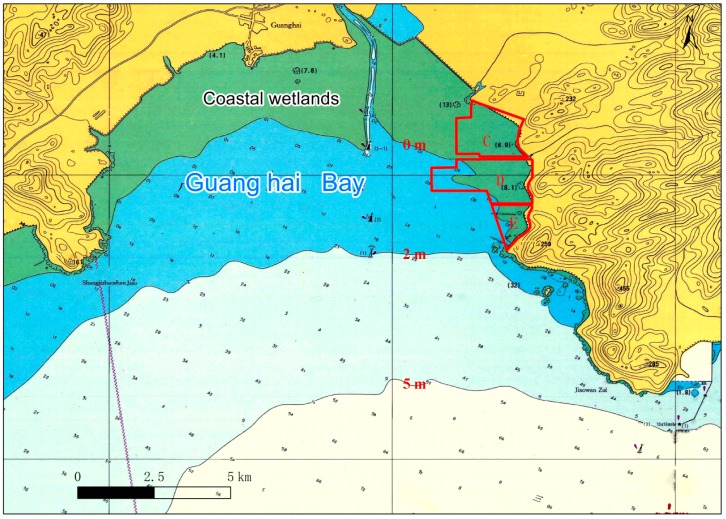
Schematic occupancy map of coastal wetlands through engineering construction.

**Figure 4 ijerph-16-04303-f004:**
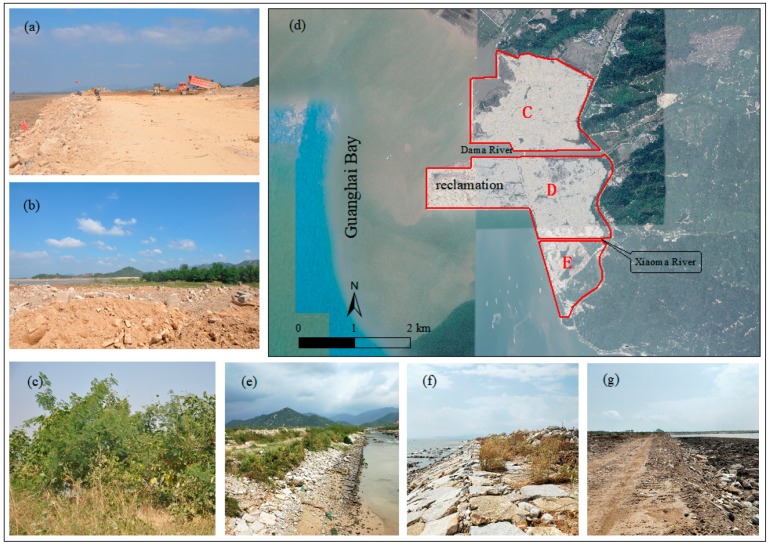
(**a**) (**b**) Soil in the reclamation area; (**c**) Plants in the reclamation area; (**d**) Reclamation layout; (**e**) Sloping shoreline in area C; (**f**) Sloping shoreline in area D; and (**g**) Sloping shoreline in area E.

**Figure 5 ijerph-16-04303-f005:**
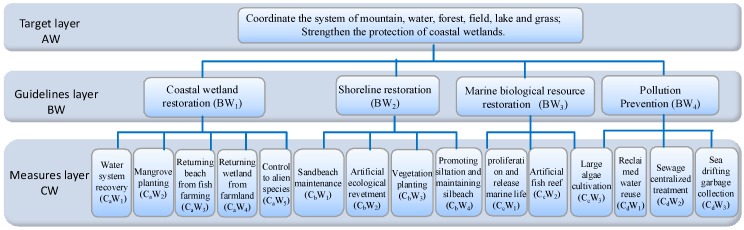
Hierarchical model of the reclamation projects and adjacent marine ecosystems. AW: target layer, BW: guidelines layer, CW: measures layer.

**Figure 6 ijerph-16-04303-f006:**
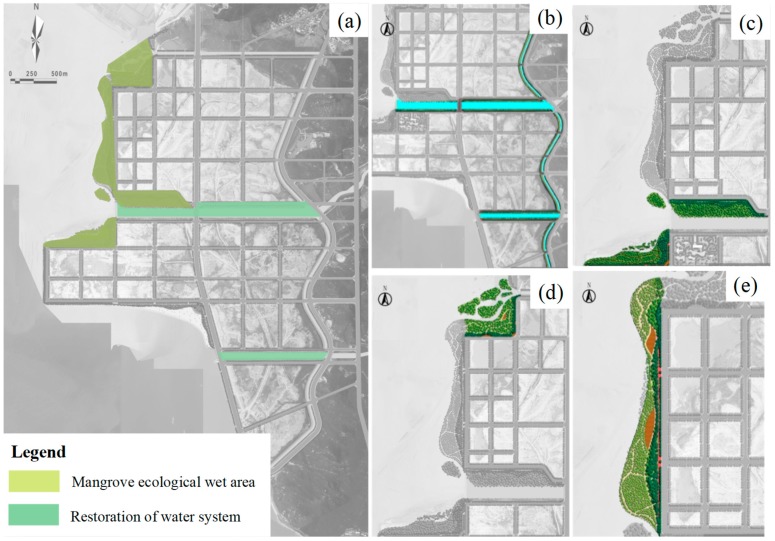
(**a**) The restoration schemes of wetland space; (**b**) Restoration of water system; (**c**) Mangrove ecological wetland area outside the Dama River; (**d**) Mangrove ecological wetland area at north side of area C and (**e**) Mangrove ecological wetland area at west side of area C.

**Figure 7 ijerph-16-04303-f007:**
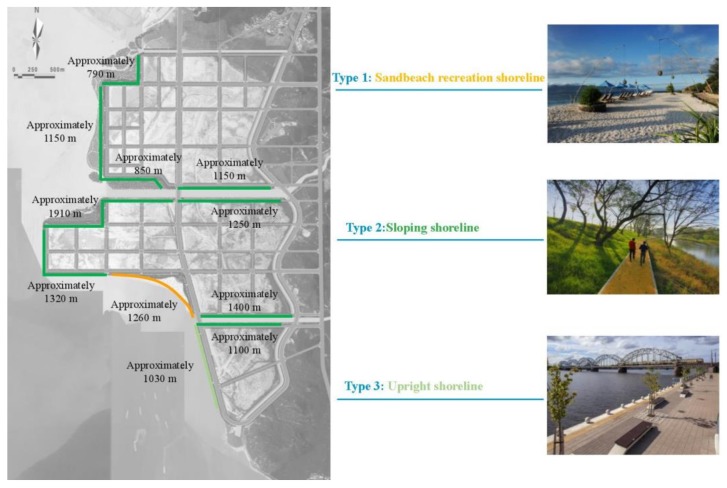
The ecological-shoreline-repair scheme.

**Figure 8 ijerph-16-04303-f008:**
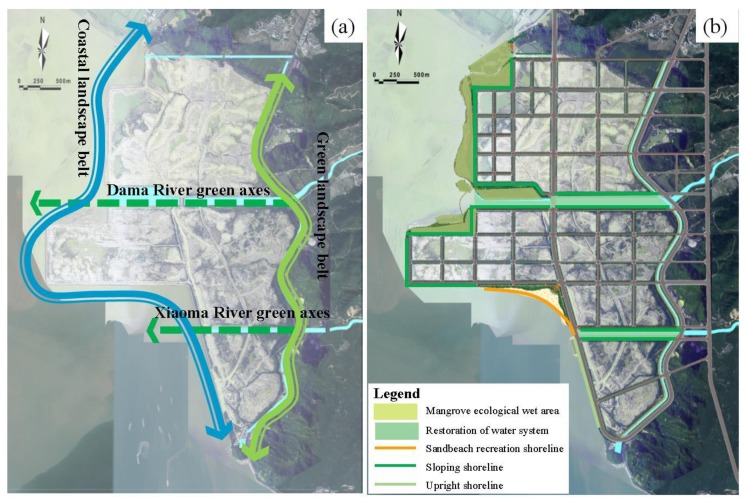
Restoration space pattern: (**a**) landscape structure and (**b**) ecological restoration.

**Figure 9 ijerph-16-04303-f009:**
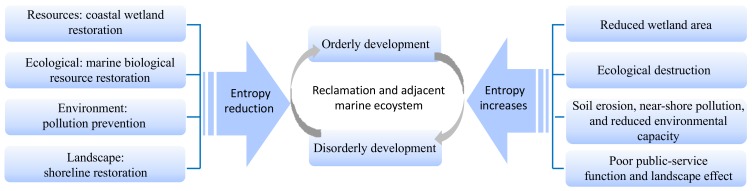
Reclamation projects and adjacent marine ecosystems development mechanism.

**Table 1 ijerph-16-04303-t001:** Analytical method with two-two comparison scale [23].

Element	Scaling	Value Rule (a Factor in the Above Layer is the Criterion, and at the Current Level, Factor *i* is Compared with Factor *j*)
*a_ij_*	1	Equally important
	3	*i* is slightly more important than *j*
	5	*i* is more important than *j*
	7	*i* is more important than *j*
	9	*i* is extremely more important than *j*
	2, 4, 6, 8	Comparison between the importance of the two factors *i* and *j* is in the middle of the above results
*a_ji_*	Reciprocal	Comparison between the importance of factors *i* and *j* is the reciprocal of the comparison between their importance

**Table 2 ijerph-16-04303-t002:** Average random consistency indicator, *RI.*

*n*	1	2	3	4	5	6	7	8	9
*RI*	0	0	0.58	0.94	1.12	1.24	1.32	1.41	1.45

**Table 3 ijerph-16-04303-t003:** Weight vector calculations and one-time test results.

Matrix	*n*	Hierarchical Single-Sorted Weight Vector (*W*)	Maximum Eigenvalue (λmax)	Average Random Consistency Indicator (*RI*)	Consistency Indicator (*CI*)	Consistency Ratio (*CR*)	Acceptable Consistency
AW	4	(0.4824, 0.2718, 0.0883, 0.1575)	4.015	1.12	0.005	0.004	Yes
BW_1_	5	(0.4461, 0.2864, 0.1567, 0.0716, 0.0392)	5.050	1.12	0.042	0.034	Yes
BW_2_	4	(0.4758, 0.2884, 0.1544, 0.0813)	4.021	0.94	0.007	0.007	Yes
BW_3_	3	(0.5390, 0.2973, 0.1638)	3.009	0.58	0.004	0.008	Yes
BW_4_	3	(0.5390, 0.2973, 0.1638)	3.009	0.58	0.004	0.008	Yes

AW: target layer, BW: guidelines layer.

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
