# Peer review of "Dissipation Theory-Based Ecological Protection and Restoration Scheme Construction for Reclamation Projects and Adjacent Marine Ecosystems"

_ijerph, 2019, doi:10.3390/ijerph16214303_

Round 1
Reviewer 1 Report
Dear Authors, The manuscript is well organized. Kindly review the manuscript for minor grammatical errors. As discussed by the authors, this manuscript describes the qualitative aspects of the entropy model of a reclamation project. However, the authors need to include information on the usefulness and practical application of this model. How will this model be validated? Explain the usefulness of this study for the construction of the qualitative analyses model. What information from this study can be used for the quantitative process. Give examples. What is the application of this qualitative study in designing a reclamation project for this area? What aspects of this study can be practically implemented ? What is economically feasible? What are the limitations? Kindly address minor revision to the following lines.Pg 2, line 76, explain riprap slope and include a reference to Fig 4 d. Pg 4, line 116, '.. system will change..'
Reviewer 2 Report
This manuscript addresses Guanghai Bay, Jiangmen City, Guangdong Province, a typical China's coastal areas, to simulate with a reduction of entropy flow, which is still relatively interesting and is of great current concern to those dealing with GIS data associated with ocean quantity/quality from coastal zone scales. I found the manuscript sufficiently original and interesting to warrant future publication after it is revised. In general, the manuscript is well written. The structure and presentation of the paper is clear. Case study is well-executed and a good demonstration of the viability and usefulness of the model for watershed planning. Limitations of model and needed future developments are clearly addressed. Article is clearly written and well organized. Literature review is extensive and references are appropriate. However, the novelty of the work is not present in a good way. The authors need to modify the introduction part and highlight the novelty, as well as the objective of the study. There is one specific question: How the authors get the dissipative structure from the conversions from establishment of entropy model in coastal landscapes in response to urbanization in coastal areas (line 82)? Why do you want to study equilibrium and/or non-equilibrium in a dissipative structure? How to measure an entropy model in an ocean? You need more explanation for our international readers.
Introduction
Please add your hypothesis and your objective of this study in land uses. It needs to be more clearly defined and how the results may aid research in a wider context in your model.
Materials and Methods
Your modelling in coastal areas, open water areas, and urban areas, should be considered to measure ecosystems from the perspective of global warming for each connected coastal corridors and waterbodies in a substantive dynamic model. Any model could be involved in these aforementioned data? Any anthropogenic data and/or natural process could be used? How to measure land-use change in your individual land conversion is crucial to calculate in your landscape metric analyses. If you consider four class-level metrics (i.e., metrics that apply to land use classes), then, as I said I am interested on your process simulation: 3.1.2. Increased environmental entropy: soil erosion, near-shore pollution, and reduced environmental capacity (line 195-196). How to measure negative entropy? How to measure natural-science measurement associated with social science (i.e., AHP measure)? How many experts measured by your Analytic Hierarchy Process (AHP, now I see your model in Fig. 5)? Please indicate your process in your methods.
Who detected “coastal wetland restoration > shoreline restoration > pollution prevention > marine biological resource restoration. Please indicate your reliability and validity. How to measure environmental negative entropy inflow. Why this flow? Any conflicts with your arguments? Any limitation in your model?
Case Study
1) Why “environmental negative entropy inflow” (line 328)? What is the difference with “bio-ecological negative entropy inflow” (line 338). Please define.
2) Since “negative entropy reduction calculated through AHP” (line 350) in this case, but I do not know how you measured and how to combine negative entropy reduction (natural science) with AHP (social science) by decision makers in this area. How to get Figure 9? Any restoration process is “decrease of entropy” (line 370)? I hope that you provide a straightforward approach with a useful tool to prove your statements and figures, and for these reasons I feel that your subject matter would be of interest to our readers.
Reviewer 3 Report
The paper is well designed and structured. There are some minor revisions that need to be considered:
To list a couple of consecutive references in the text, there is no need to put the number of each and every one of those references. e.g., line 36 can be revised as [2-9]. Line 74, when introducing different areas of A to E, please refer to Figure 3 to have a better understanding of their location in the case area or show them in Figure 1, since the whole paragraph is referring to Figure 1. Line 76, what is hm2? Line 88, how a system cannot be equilibrium? Line 120, most probably it should be segment CE instead of DE. Line 125, it seems that the caption of Figure 2 is not complete. Growth and decline in what? Line 146, in the text where referring to Table 1, a reference is also introduced. If Table 1 is exactly extract from this reference, I would suggest to include the reference in the caption of this table as well. Line 163, N should be changed to n in Table 2. Lines 209-210, There is no need to repeat the whole sentence of "According to the results of a July 2017 ... the Ocean University of China" since it has been mentioned/introduced in previously in the text. References 3 and 4 (for instance), is it "Department of Natural Resources" or "Ministry of Natural Resources"? It should be consistent among all references. Most of the references are in Chinese which makes it difficult for non-Chinese people to check/read/find them. Even though the title of the research is translated in English in some cases, it is still impossible to find them. For references 14 and 15 it seems that the first name of the authors is mentioned in the text (lines 50-52) which need to be revised to the last names.Author Response
Please see attachment

Reviewer 4 Report
The first subchapter of the Results section (3.1) provides some general or qualitative information about the study area-specific aspects of increase in the entropy, while in the Methods section, only the entropy model development is deteiled. It should be considered if some of these information might be included in the Study area, or in the Discussion sections.
In line 148, the "Consistency test" subchapter is numbered with 3, while it should have 2.2.3.
Another careful English and spelling check is required.
Round 2
Reviewer 1 Report
Thank you submitting the revised manuscript. You have made significant improvement in the overall content and quality. Kindly find my recommendation for the figures below.
Figure 3: The text in figure 3 is not legible. The figure will need a higher resolution or a replacement with a different figure. Figure 4 d: The text in green font is not legible Figure 5 and 9 : Recommend using different font and not using bold textAuthor Response
Response:
Thank you for your thoughtful comments, which have helped us to produce a paper that will appeal to a wider readership.
Figure 3: was replaced as a clearer Figure and its details were revised.
Figure 4d: green font was replaced as black font.
Figure 5 and 9: Times New Roman font was replaced as Yu Gothic UI font.
In addition, we have modified other oversights, please see the red font section of the article.

Reviewer 2 Report
Reasonably well written. I may encourage the authors to expand on stakeholder engagement (i.e., through AHP), planning, and management of these ocean resources, and green energy for your topics in the nearby future.
Please double check that your manuscript is correctly formatted and all address minor revision in your statements, such as page 10, line 308, please correct your superscript from your statement, “the 270000 m2, and the planting density is 1 plant/m2”.
Author Response
Response:
Thank you for your thoughtful comments, which have helped us to produce a paper that will appeal to a wider readership.
We have revised this oversight. For details, please see Lines 310 on Page 10. In addition, we have modified other oversights, please see the red font section of the article.
Inevitably, we will expand on stakeholder engagement, planning, and management in the nearby future.
